# *Pestalotiopsis* Diversity: Species, Dispositions, Secondary Metabolites, and Bioactivities

**DOI:** 10.3390/molecules27228088

**Published:** 2022-11-21

**Authors:** Chu Wu, Yun Wang, Yujie Yang

**Affiliations:** 1College of Horticulture & Gardening, Yangtze University, Jingzhou 434025, China; 2College of Life Sciences, Yangtze University, Jingzhou 434025, China

**Keywords:** *Pestalotiopsis* genus, secondary metabolites, biosynthesis, bioactivity

## Abstract

*Pestalotiopsis* species have gained attention thanks to their structurally complex and biologically active secondary metabolites. In past decades, several new secondary metabolites were isolated and identified. Their bioactivities were tested, including anticancer, antifungal, antibacterial, and nematicidal activity. Since the previous review published in 2014, new secondary metabolites were isolated and identified from *Pestalotiopsis* species and unidentified strains. This review gathered published articles from 2014 to 2021 and focused on 239 new secondary metabolites and their bioactivities. To date, 384 *Pestalotiopsis* species have been discovered in diverse ecological habitats, with the majority of them unstudied. Some may contain secondary metabolites with unique bioactivities that might benefit pharmacology.

## 1. Introduction

As human society entered a new century, many new problems have to be faced, such as global warming, public health, and food crisis. Especially, novel coronavirus burst in 2020 spring caused a severe effect on global public health and the world economy. More effective novel medications will be investigated to respond to emerging public health challenges. Many bioactive components were already isolated and identified from plants, animals, bacteria, and fungi. Because of the great numbers of bacteria and fungi and their various habitats, they are important sources of bioactive components. Subramanian and Marudhamuthus (2020) [1] isolated and identified the endophytic bacteria, such as *Bacillus flexus* (DMTMMB08), *Bacillus licheniforms* (DMTMMB10), and *Oceanobacillus picturae* (DMTMMB24) from marine macroalgae *Sargassum polycystum* and *Acanthaphora specifera* in the benthic region of the Gulf of Mannar, and found that they are taxol-producing. The endophytic fungus *Taxomyces andreanae* was isolated from the outer bark of *Taxus brevifolia* and was first found to have the ability to produce taxol in a culture medium, at approximately 24–25 ng/L [2]. Since then, a great number of taxol-producing fungi, acting as endophytic fungi, have been isolated and identified, such as the endophytic fungus *Chaetomella raphigera* from a medicinal plant, *Terminalia arjuna* [3], the endophytic fungus *Epicoccum nigrum* TXB502 [4], and the endophytic fungus *Penicillium polonicum* from *Ginko biloba* [5]. Among them, *Pestalotiopsis* species have been widely studied. The fungal genus *Pestalotiopsis* was first established by Steyaert R. L. [6]. Since then, many *Pestalotiopsis* species have been isolated and identified. To date, 384 *Pestalotiopsis* species are listed in the Index Fungorum (http://www.indexfungorum.org/Names/Names.asp, assessed on 1 August, 2022). All the described species in the *Pestalotiopsis* genus are differentiated primarily on morphological characteristics of conidia, conidiogenesis, teleomorph, and host associations. In addition, the presence or absence of basal and apical appendages can be used as additional taxonomic characters for identifying *Pestalotiopsis* species. They are widely distributed in tropical and temperate regions [7,8,9,10]. As early as 1996, taxol was first isolated and identified in *Pestalotiopsis microspora*, an endophytic fungus of *Taxus wallachiana* [11]. Besides *Pestalotiopsis microspora* [12,13], some *Pestalotiopsis* species were also found to produce taxol, such as *Pestalotiopsis mangiferae* [14], *Pestalotiopsis pauciseta* [15], *Pestalotiopsis breviseta* [16,17], *Pestalotiopsis terminaliae* [18], and *Pestalotiopsis hainanensis* [19]. *Pestalotiopsis microspora* was found to produce between 50 and 1487 ng/L taxol, indicating that taxol production could be achieved at a higher concentration; however, the production was found unstable due to different fungal strains of *P. microspora* [20].

Since taxol discovery in *P. microspora*, several other secondary metabolites were isolated and identified from *Pestalotiopsis* species, and their bioactivities were tested. These secondary metabolites possess several bioactivities, such as anticancer, antifungal, antibacterial, antivirus, and insecticide. Yang XL et al. [21] and Xu J et al. [10] summarized secondary metabolites from *Pestalotiopsis* species. Although Helaly et al. [22] and Becker and Stadler [23] recently summarized secondary metabolites from the order Xylariales, to which the family Sporocadaceae belongs (*Pestalotiopsis* is a genus of ascomycete in the family Sporocadaceae), they did not focus on these secondary metabolites from the genus *Pestalotiopsis*. In this review, we summarize recent advances in secondary metabolites isolated and identified from *Pestalotiopsis* species. Our goal is to encourage the use of these secondary metabolites and the discovery of new *Pestalotiopsis* metabolites.

## 2. Habitat and Functional Diversity of *Pestalotiopsis* Species

*Pestalotiopsis* species reside in various habitats, including oceans, rivers, lakes, air, soil, and different plant tissues. For instance, *Pestalotiopsis submerses*, as a root endophytic fungus, were identified in the roots of plants residing in wetlands near ravine areas with elevations of 1150 and 1775 m [24]. The fungus lives in the roots of *Equisetum* sp., fern, and *Lyonia ovalifolia*, yet the occurrence frequency is low (12.5%). *Pestalotiopsis* species were isolated in the outside air of all four child daycare centers tested by Aydogdu and Asan [25]; however, they did not discover these fungi in the indoor air of these child daycare centers. Some *Pestalotiopsis* species come from the marine environment, such as *Pestalotiopsis neglecta* [26,27], *Pestalotiopsis heterocornis* [28,29,30,31], *Pestalotiopsis vaccinii* [32], *Pestalotiopsis sydowiana* [33], and other *Pestalotiopsis* sp. [34,35,36,37]. *Pestalotiopsis papuana* CBS 331.96 and *Pestalotiopsis humus* CBS 336.97, on the other hand, were discovered in soil [8]. Astonishingly, *Pestalotiopsis hainanensis* was identified from the dermatitic scurf of a giant panda (*Ailuropoda melanoleuca*) [19] and *Pestalotiopsis* sp. HC02 resides in *Chondracris rosee* gut [38]. *Pestalotiopsis* ssp. was also identified as an entomopathogenic fungus on *Hemiberlesia pitysophila*, an extremely harmful exotic insect in *Pinus* forests [39]. Thus, the isolate is promising for the biocontrol of *H. pitysophila*. In addition, *Pestalotiopsis* species might be related to human diseases. For example, *Pestalotiopsis clavispora* was identified from a patient’s cornea with recurrent keratitis [40].

In their respective habitats, *Pestalotiopsis* species show different ecological functions. *Pestalotiopsis* species, as plant endophytic or saprophytic fungi, reside in other plant tissues, such as bark [41], stems [42], twigs [43], roots [24], leaves [44,45,46,47,48], flowers [49,50], and fruits [51]. As saprophytic fungi, they cause various plant diseases, such as blight diseases in leaves and twigs [43,50,52,53,54,55], leaf necrosis [44], leaf spots [46], fruit rot [51], dry flower disease [56], dieback disease [57,58,59], canker [58,60,61], and even postharvest diseases [62]. Sometimes, *Pestalotiopsis* species cause destructive diseases. For example, *Pestalotiopsis samarangensis* was harmful to *Syzygium samarangense* in Thailand [63]. Some *Pestalotiopsis* species act as an endophyte or a saprobe in different plant species. For example, *P. microspora* causes leaf spots in blueberry (V*accinium corymbosum* L.) [46], leaf blight of loquat (*Eriobotrya japonica*) [64], and leaf blight of Japanese yew (*Taxus cuspidate*) [65]; however, the fungus more often acts as an endophyte [66,67,68]. Therefore, *Pestalotiopsis* species play a variety of ecological roles.

## 3. New Secondary Metabolites in *Pestalotiopsis* Species

In 2012, it was estimated that 196 secondary metabolites had been encountered in the *Pestalotiopsis* genus [21]. New secondary metabolites were isolated and identified in the recent decade, and their bioactivities were tested (Table 1). We compared the compounds in the literature published after 2014 with those listed by Yang X-L et al. [21] and Xu J et al. [10] and came up with the following list of novel compounds listed below.

### 3.1. Pestalotiopsis diploclisia

The marine fungus *P. diploclisia* (BCC 35283) produces two new hydroquinones bearing a 1,3-enyne moiety, pestalotioquinols G (**1**) and H (**2**) [73]. *P. microspora* has pestalotioquinol A (**3**) and B (**4**), and the two compounds are neuroprotective [121]. Furthermore, 1–3 μM pretreatment of pestalotioquinols A (**3**) and B (**4**) rescued nerve growth factor-differentiated neuronal PC12 cells from peroxynitrite-induced cytotoxicity, and their protective activity was sustained after removing each compound from the medium; thus, the two compounds (**3** and **4**) exhibited relatively high neuroprotective effects. In addition, Pestalotioquinol A (**3**) displayed antimalarial activity against *Plasmodium falciparum* K1 with an IC_50_ value of 19.0 μM. In comparison, pestalotioquinol G (**1**) showed weak cytotoxic activity against Vero cell lines with an IC_50_ value of 47.9 μM [73].



*P. diploclisia* also produces the compound scylatone (**5**) [73]. Scylatone (**5**) is one of the melanin biosynthesis intermediates. *P. microspora* and *Pestalotiopsis fici* were reported to produce melanin pigment [13,84,90,114,116]. Melanin biosynthesis is complex in fungi, and some fungi have more than one biosynthesis pathway for melanins [185]. Melanization in mycelia and appressoria plays crucial roles in the protection of pathogens from antibiotic stressors and the pathogenicity or interaction with host plants [185,186], and melanin is essential not only for the protection of spores from biotic and abiotic stresses but also structural spore development [84]. Thus, scylatone and melanin are related to the infection of host plants.



### 3.2. Pestalotiopsis disseminate

*P. disseminate* produces disseminins A-E (**6**–**10**) and spiciferones D (**11**) and E (**12**) [75]. Disseminins A-E (**6**–**10**) showed no activity against *Aspergillus flavus*, *Escherichia coli*, or *Candida albicans* at 100 μg/disk [75]. These compounds are synthesized via the pathway shown in Figure 1 [75]:

Spiciferone A (**13**) and other congeners (**14**–**21**) were also isolated and identified in *Cadophora luteo-olivacea* collected from Port Lockroy on the Antarctic peninsula, endophytic fungus *Phoma betae* collected from desert plants in West China [187,188], and *Cochlioholus spicifer* [12]. Spiciferone A (**13**) is a major phytotoxin to cotyledons of wheat (*Triticum aestivum* L. cv. Ushio-komugi), and spiciferone C (**20**) was less phytotoxic to wheat cotyledons than spiciferone A (**13**), while spiciferone B (**19**) was not phytotoxic, and dihydrospiciferone A (**21**) was as active as spiciferone A (**13**) [12]. Hwang et al. [75] did not observe that spiciferones D (**9**) and E (**10**) showed activity against *Staphylococcus aureus*, *Bacillus subtilis*, *E. coli*, and *C. albicans* at 50 μg/disk. Spiciferone A (**13**), spiciferol A (**14**), dihydrospiciferol A (**15**), spiciferone F (**16**), and dihydrospiciferone A (**21**) showed no activity against methicillin-resistant *S. aureus* (MRSA), vancomycin-resistant *Enterococcus faecalis* (VRE), *B. subtilis*, *Acinetobacter baumannii*, *Pseudomonas aeruginosa*, *Klebsiella pneumoniae*, *C. neoformans*, and *C. albicans*. These five compounds also showed no cytotoxicity against LOX IMVI (melanoma) and SF-295 (glioblastoma) human cancer cell lines [187].



### 3.3. Pestalotiopsis fici

*P. fici*, *Pestalotiopsis guepinii*, and *Pestalotiopsis theae* produce ficipyrone A (**22**) and pestheic acid (i.e., dihidromaldoxin) (**23**) [78,82,98,166]. Xu X et al. [82] proposed a biosynthesis pathway of pestheic acid. The two compounds showed moderate inhibition of respiratory syncytial virus (IC_50_ values ranging from 45.00 ± 0.98 to 259.23 ± 2.36 µM) [166]. Sousa et al. [98] assessed the cytotoxic, cytostatic, and genotoxic effects of pestheic acid (**23**) in a gastric adenocarcinoma cell line (PG100). Their results showed a decrease in clonogenic survival. Pestheic acid (**23**) also significantly increased both micronucleus and nucleoplasmic bridge frequency. However, they observed no changes in cell cycle kinetics or apoptosis induction. Thus, they considered that pestheic acid (**23**) was not an active anticancer compound under these conditions, because the minimal inhibitory concentration was high.



Pestaloficins (**24**–**28**) were isolated and identified in *P. fici* and other *Pestalotiopsis* sp. [88,189]. In addition, Guo and Zou [190] discovered pestaloficin C (**26**) in the plant endophytic fungus *Monosporascus eutypoides*. However, none of the studies focused on the bioactivities of these substances, so their bioactivities need to be tested.

### 3.4. Pestalotiopsis foedan

(−)-(4*S*,8*S*)-foedanolide (**29**) and (+)-(4*R*,8*R*)-foedanolide (**30**), a pair of new spiro-γ-lactone enantiomers, were isolated from the fermentation broth of the plant endophytic fungus *P. foedan* [94]. The two compounds showed moderate activities against HeLa cells, lung adenocarcinoma cell line A-549, U-251, HepG2, and MCF-7 tumor cell lines. The IC_50_ values of (−)-(4*S*, 8*S*)-foedanolide (**29**) against these tumor cell lines were 15.8, 296.0, 159.0, 22.8, and 70.2 μg/mL, respectively; the IC_50_ values of (+)-(4*R*,8*R*)-foedanolide (**30**) against these tumor cell lines were 5.4, 67.9, 53.0, 19.0, and 20.8 μg/mL, respectively [94].

Yang XL et al. [21] and Xu J et al. [10] introduced some sesquiterpenes, diterpenes, and triterpenes, but not monoterpenes. A new monoterpene lactone, (1*R*,4*R*,5*R*,8*S*)-8-hydroxy-4,8-dimethyl-2-oxabicyclo [3.3.1]nonan-3-one (**31**), and (2*R*)-2-[(1*R*)-4-methylcyclohex-3-en-1-yl]propanoic acid (**32**), were isolated from the liquid culture of the plant endophytic fungus *P. foedan* [95]. Both compounds (**31** and **32**) exhibited strong antifungal activities against the two pathogens, *Botrytis cinerea* and *Phytophthora nicotianae*, with MIC values of 3.1 and 6.3 μg/mL, respectively, which are comparable to those of the known antifungal drug ketoconazole. The compound (**32**) also showed modest antifungal activity against *C. albicans* with a MIC value of 50 μg/mL [95].



### 3.5. Pestalotiopsis guepinii

*P. guepinii* is the fungal causal pathogen of hazelnut (*Coryus avellana*) in Turkey. *P. guepinii* produced phytotoxic α-pyrones, including 6-(1-hydroxypentyl)-4-methoxy-pyran-2-one (**33**), derivatives (**34**, **35**, **37**, **38**) of the compound **33**, and 6-pentyl-4-methoxy-pyran-2-one (**36**) [99]. None of these compounds (**33**–**38**) showed antibiotic activities against *B. subtilis* and *Geotrichum candidum* when tested up to 100 μg per diskette [99].



### 3.6. Pestalotiopsis heterocornis

Heterocornols A-L (**39**–**50**), methyl-(2-formyl-3-hydroxyphenyl) propanoate (**51**), cladoacetal A (**52**), xylarinol A (**53**), agropyrenol (**54**), vaccinol G (**55**), (*R*)-3-hydroxy-1-[(*R*)-4-hydroxy-1,3-dihydroisobenzofuran-1-yl]butan-2-one (**56**), and (*R*)-3-hydroxy-1-[(*S*)-4-hydroxy-1,3-dihydroisobenzo-furan-1-yl]butan-2-one (**57**) were isolated and identified in the marine sponge-derived *P. heterocornis* [29]. Heterocornols A-C (**39**–**41**), Heterocornols F-H (**44**–**46**), methyl-(2-formyl-3-hydroxyphenyl) propanoate (**51**), agropyrenol (**54**), and vaccinol G (**55**) showed cytotoxic activities against four human cancer cell lines with IC_50_ values of 15–100 μM. These also showed antibacterial activities against Gram-positive bacteria *S. aureus* and *B. subtilis* with MIC values ranging from 25 to 100 mg/mL [30]. In addition, these compounds, heterocornol C (**41**), heterocornol G (**45**), agropyrenol (**54**), and vaccinol G (**55**), exhibited weak antifungal activities against *Candida parapsilosis* and *C. neoformans* with MIC values of 100 μg/mL [30].



Lei H et al. [28] also isolated and identified heterocornols M and N (**58**, **59**) and a pair of epimers, heterocornols O and P (**60**, **61**), in *P. heterocornis.* The four compounds (**58**–**61**) showed cytotoxic activities against four human cancer cell lines (BGC-823, Ichikawa, HepG2, 7860) with IC_50_ values of 20.4–94.2 µM.



In *P. heterocornis*, pestaloisocoumarins A and B (**62**, **63**), gamahorin (**64**), one sesquiterpenoid degradation product, isopolisin B (**65**), and one furan derivative, pestalotiol A (**66**), were also isolated and identified [30]. Pestaloisocoumarins A and B (**62**, **63**) and gamahorin (**64**) exhibited antibacterial activities against Gram-positive bacteria *S. aureus* and *B. subtilis* with MIC values ranging from 25 to 100 µg/mL. The three isocoumarins (**62**, **63**, **64**) showed weak antifungal activities against *C. albicans*, *Candidad parapsilosis*, and *C. neoformans* with MIC values of 100 µg/mL. Natural isocoumarins were studied in terms of sources, structural styles, biosynthesis, and biological activity by Shabir et al. [191] and Noor et al. [192], who also gave some relevant information. A mixture of pestalalactone atropisomer A/B (**67**, **68**) showed moderate cytotoxic activities against four human cancer cell lines (BGC-823, H460, PC-3, SMMC-7721) with IC_50_ values 0f 6.8–87.7 μM, while compounds **62**–**66** did not exhibit an obvious inhibition effect against the cancer lines at 100 μM [30].



### 3.7. Pestalotiopsis humus

Pestynol (**69**) and pestiocandin (**70**) were isolated and identified in *P. humus* FKI-7473 using multidrug-sensitive yeast [102,103]. Pestynol (**69**) was weakly cytotoxic against three floating cell lines (Jurkat, HL60, and THP-1 cells) and two adherent cell lines (HT29 and A549 cells), with IC_50_ values of 84, 19, 61, 83, and 92 μM, respectively, and the other three adherent cell lines (HeLa S3, H1299, and Panc1 cells) were not affected at 100 μM of pestynol (**69**) [102]. Pestiocandin (**70**) showed moderate or weak growth inhibition against Gram-positive bacteria (*S. aureus*, *B. subtilis*, *Micrococcus luteus*, and *Mycobacterium smegmatis*), Gram-negative bacteria (*Xanthomonas oryzae* pv. *oryzae* and *Proteus vulgaris*), yeasts (*Saccharomyces cerevisiae* BY25929 and *S. cerevisiae* 12geneΔ0HSR-iERG6), and a filamentous fungus (*Mucor racemosus*), with MIC values of 8–256 μg/mL [103]. The compound (**70**) did not exhibit effects on Gram-negative bacteria (*B. subtilis*, *K. pneumoniae*, and *P. aeruginosa*) and yeasts (*C. albicans*, *S. cerevisiae* KF237, and *S. cerevisiae* BY4741), with MIC values of more than 256 μg/mL.



### 3.8. Pestalotiopsis karstenii

Two oxysporone derivatives, pestalrone A (**71**) and pestalrone B (**72**), were isolated and characterized in the endophytic plant fungus *P. karstenii* isolated from stems of *Camellia sasanqua* and foliar endophytic fungus *Pestalotiopsis zonata* [42,184]. Pestalrone A (**71**) had no inhibitory effects on the five human tumor cell lines (HeLa, U-251, A549, HepG2, and MCF-7). At the same time, pestalrone B (**72**) showed significant activities against the three human tumor cell lines (HeLa, HepG2, and U-251) with IC50 values of 12.6, 31.7, and 5.4 µg/mL, respectively. However, this compound (**72**) did not show noticeable cytotoxic activities against cell lines A549 and MCF-7 [42]. In addition, pestazonatic acid (**73**) was isolated and identified in *P. zonata* [184], yet its bioactivity was not evaluated.



### 3.9. Pestalotiopsis mangiferae

A phenolic compound, 4-(2,4,7-trioxa-bicyclo [4.1.0]heptan-3-yl) phenol (**74**), was isolated and characterized from the endophytic fungus *P. mangiferae* associated with *Mangifera indica* [107]. The compound (**74**) showed appreciable antibacterial and antifungal activities against *B. subtilis*, *K. pneumoniae, C. albicans* (MIC of 0.039 mg/mL), *B. subtilis,* and *M. luteus* (MIC of 1.25 mg/mL) followed by *P. aeruginosa* (MIC of 5.0 mg/mL).

### 3.10. Pestalotiopsis microspora

Yang XL et al. [21] and Xu J et al. [10] introduced novel sesquiterpenes. Three new sesquiterpenes, (+)-dendocarbin L (**75**), (+)-sydonic acid (**76**), and (+)-sydowic acid (**77**), were isolated from the mycelium of the endophytic fungus *P. microspora* associated with the stem of *Artocarpus heterophyllus* [68]. Their cytotoxicity was evaluated on murine leukemia P-388 cells, with IC_50_ values of 18.78, 20.30, and 2.56 μg/mL for (+)-dendocarbin L (**75**), (+)-sydonic acid (**76**), and (+)-sydowic acid (**77**), respectively. Thus, among the three compounds (**75**, **76**, **77**), (+)-sydowic acid (**77**) exhibits the most potential to inhibit murine leukemia P-388 cells.



Yang et al. [21] presented pestalotioprolides A and B. New analogs were isolated and identified as well. Pestalotioprolides C-H (**78**–**83**) and 7-*O*-methylnigrosporolide (**84**) were isolated from a mangrove-derived endophytic fungus *P. microspora* [118]. The four compounds, 7-*O*-methylnigrosporolide (**84**), pestalotioprolide D (**79**), pestalotioprolide E (**80**), and pestalotioprolide F (**81**), showed significant cytotoxicity against the murine lymphoma cell line L5178Y with IC50 values of 0.7, 5.6, 3.4, and 3.9 μM, respectively, while pestalotioprolide E (**80**) exhibited potent activity against the human ovarian cancer cell line A2780 with an IC_50_ value of 1.2 μM. Interestingly, co-culture of *P. microspora* with *Streptomyces lividans* caused an approximately 10-fold enhanced accumulation of pestalotioprolide E (**80**) and pestalotioprolide F (**81**), compared to axenic fungal controls [118]. The enhanced accumulation is beneficial for the formation of pestalotioprolide E (**80**) and pestalotioprolide F (**81**), and attention should be directed to associated molecular pathways.

In addition, nigrosporolide (**85**) and 4,7-dihydroxy-13-tetradeca-2,5,8-trienolide (**86**) were isolated in *P. microspora* and the mold *Nigrospora sphaerica* [118,193]. The two compounds (**85** and **86**) showed weak bioactivities against lymphoma cell line L5178Y with an IC_50_ value of 21 μM, while they exhibited no biological activities against ovarian cancer cell line A2780 with an IC_50_ value of more than 40 μM [118]. Furthermore, Harwooda et al. [193] reported that nigrosporolide (**85**) caused 100% inhibition in the growth of etiolated wheat coleoptile sections at 10^−3^ M; however, it showed no effect at lower concentrations. Since auxins have a significant impact on plant development, nigrosporolide (**85**) could be analyzed as a new component in the auxin signaling pathway, such as the TOR kinase found via rapamycin produced by the soil bacterium *Streptomyces hygroscopicus* in the Easter Island [194,195].



Pitholide B (**87**), pitholide D (**88**), and pitholide E (**89**) were isolated and identified in *P. microspora* and *Pithomyces* sp. [67,196]. However, Pitholide E (**89**) did not exhibit any significant antifungal activity against *Cladosporium cladosporioides*, while Pitholide B (**87**) and Pitholide D (**88**) were not analyzed for their bioactivity [67]. In addition, Pitholides A (**90**) and C (**91**) were isolated from *Pithomyces* sp. derived from the marine tunicate *Oxycorynia fascicularis*, and their bioactivities were also not evaluated [196].



*P. microspora* SC3082 was derived from the tropical tree *Scaevola taccada*, and a few new compounds (**92**–**97**) were isolated and characterized in the strain, including microsporaline A-D (**92**–**95**) [66]. Microsporalines B and C (**93**, **94**) and gamahorin (**96**) displayed moderate antifungal activities against *C. albicans* (ATCC 10321) with MIC values of 25.0, 25.0, and 12.5 μg/mL, respectively, while microsporaline A and D (**92**, **95**) and 8-acetoxy pestalopyrone (**97**) did not show bioactivities against *C. albicans* (ATCC 10321) with MIC values of > 100 μg/mL.



Yang XL et al. [21] introduced ambuic acid and several of its derivatives, and more compounds have been discovered since then. For example, three new ambuic acid derivatives (**98**–**100**) were isolated and characterized in *P. microspora* found from the leaves of *Corylus chinensis* [122]. Among them, microsporol A (**98**) and microsporol C (**100**) exhibited moderately inhibitory effects on human 5-lipoxygenase (5-LOX), by 48.32% and 58.72%, respectively.



### 3.11. Pestalotiopsis neglecta

Four new ambuic acid derivatives (**101**–**104**) were isolated from the solid culture of *P. neglecta* [123]. In the nitric oxide (NO) inhibition assay, the derivative **104** exhibited weak inhibitory activity against the NO production in the lipopolysaccharide (LPS)-induced macrophage with an IC_50_ value of 88.66 μM. Since NO shows a strong physiological function in plants and humans, the derivative **104** could explore the novel NO functions.



Pestalotiochromenoic acids A-D (**105**–**108**), as well as two novel chromone derivatives, pestalotiochromones A (**109**) and B (**110**), were discovered for the first time in the marine algae-derived fungus *P. neglecta* SCSIO41403 [125]. All of these compounds (**105**–**110**) were inactive or showed weak cytotoxicity against seven human cancer cell lines, i.e., three renal cancer cell lines (ACHN, OS-RC-2, 786-O), three leukemia cell lines (HL-60, K-562, MOLT-4), and a liver hepatocellular carcinoma cell line (HepG2) (IC_50_ > 50 μM); however, these compounds showed obvious liver X receptors (LXRs) modulatory activities in a dose-dependent manner. From the liquid cultures of *P. neglecta* SCSIO41403, three new carboxylic acid derivatives, pestallic acids F (**111**) and G (**112**), pestalotiopyrone N (**113**), and a new diphenylketone derivative, neopestalone (**114**), were obtained [27]. In addition, other known compounds were found in the strain, including sesquicaranoic acid B (**115**) and monocycloalternarene B (**116**), which were not included in the articles by Yang XL et al. [21] and Xu J et al. [10]. However, these compounds (**111**–**116**) did not exhibit biological activities against Dengue virus virulence with 10 μM, while neopestalone (**114**) exhibited obvious COX-2 inhibitory activities, with the IC_50_ value of 5.8 μM.



### 3.12. Pestalotiopsis palmarum

Four diphenyl ether derivatives, sinopestalotiollides A-D (**117**–**120**), and one natural α-pyrone product (**121**), were newly obtained from the ethyl acetate extract of the endophytic fungus *P. palmarump* isolated from the leaves of medicinal plant *Sinomenium acutum* in China [64]. Sinopestalotiollides A-C (**117**–**119**) and 5,6-dihydro-4-methoxy-6-hydroxymethyl-2H-pyran-2-one (**121**) exhibited moderate cytotoxicities against two human tumor cell lines, HeLa and HCT116, and showed weak cytotoxicities against human tumor cell line A549. IC_50_ values of sinopestalotiollide A (**117**) against HeLa, HCT116, and A549 were 18.92, 15.69, and 31.29 μM, respectively; IC_50_ values of sinopestalotiollide B (**118)** against the cell lines were 12.80, 22.67, and 44.89 μM, respectively; IC_50_ values of sinopestalotiollide C (**119)** were 14.66, 18.49, and 36.13 μM, respectively. IC_50_ values of 5,6-dihydro-4-methoxy-6-hydroxymethyl-2H-pyran-2-one (**121)** were 15.60, 24.35, and 47.82 μM, respectively.



### 3.13. Pestalotiopsis sydowiana

Xia X et al. [165] isolated and identified our penicillide derivatives (**122**–**125**) in *P. sydowiana.* The three compounds, 3′-O-methyldehydroisopenicillide (**122**) and pestalotiollide A and B (**123**, **124**), exhibited inhibitory activity against the 20S proteasome, with the IC_50_ values of 30.5, 12.4, and 18.5 μM, respectively.



### 3.14. Pestalotiopsis theae

Two new spiroketals with unique skeletons, chlorotheolides A (**126**) and B (**127**), and a new methylenesuccinic acid derivative, 1-undecene-2,3-dicarboxylic acid (**128**), were isolated and identified in the endophytic fungus *P. theae* [170]. In addition, their precursor, maldoxin (**129**), was also isolated. Chlorotheolides A (**126**) and B (**127**) could be biogenetically generated from the co-isolated 1-undecene-2,3-dicarboxylic acid (**128**) and maldoxin (**129**) via Diels-Alder reactions, as shown in Figure 2.

The hypothetical biosynthetic pathways for chlorotheolides A (**126**) and B (**127**) are shown in Figure 3.

Chlorotheolides A (**126**) and B (**127**) exhibited weak inhibitory effects on two human tumor cell lines, HeLa (cervical carcinoma) and MCF-7 (breast adenocarcinoma), with IC_50_ values of 13.3–73.2 μM, compared to the positive control cisplatin (IC_50_ values of 4.7 and 4.9 μM, respectively), and chlorotheolides B (**127**), which was found to inhibit cell viability in a time- and dose-dependent manner; however, 1-undecene-2,3-dicarboxylic acid (**128**) and maldoxin (**129**) did not display detectable activity against the tested tumor cell lines at a concentration of 20 μg/mL [170].



Liu G et al. [174] found two new humulane-derived sesquiterpenoids, pestalothenin A (**130**) and B (**131**), and one new caryophyllene-derived sesquiterpenoid (**132**) in *P. theae*. However, the three compounds (**130**–**132**) did not exhibit detectable inhibitory effects on five human tumor cell lines, A549 (human lung adenocarcinoma cell line), T24 (human bladder carcinoma cell line), HeLa (human cervical carcinoma cell line), MCF-7 (human breast cancer cell line), and HepG2 (human hepatoma cell line) at 50 μM, and neither showed antibacterial activities against *S. aureus* (CGMCC 1.2465), *B. subtilis* (ATCC 6633), *Streptococcus pneumoniae* (CGMCC 1.1692), and *B. subtilis* (CGMCC 1.2340) (MIC > 50 µg/mL) [174].

Two new caryophyllene-type sesquiterpenoids, pestathenols A (**133**) and B (**134**), and one new α-furanone, pestatheranone A (**135**), were also recently isolated and characterized in *P. theae* [173]. Pestathenol A (**133**) and pestathenol B (**134**) exhibited cytotoxicity against the HeLa tumor cell line, with IC_50_ values of 78.2 and 88.4 μmol/L, respectively. However, these sesquiterpenoids did not show cytotoxicity against tumor cell lines MCF-7, HepG2, and ACHN (human renal carcinoma cell line). In comparison, pestatheranone A (**135**) did not show detectable inhibitory effects on the cell lines tested at 100 μmol/L as well [173].



### 3.15. Pestalotiopsis uvicola

From the fungus *P. uvicola*, a new hybrid of dehydroergosterol and nitrogenous alternariol derivative, pestauvicomorpholine A (**136**), and three alternariol analogs (**137**–**139**), including a new aminated one, pestauvicolactone A (**137**), were found by [175]. Figure 4 depicts their possible biogenetic relationship. The two new compounds, pestauvicomorpholine A (**136**) and pestauvicolactone A (**137**), did not exhibit cytotoxicity against a mouse melanoma (B16-BL6) cell line at a concentration of 30 μM [175].



Endophytic *P. uvicola* was further isolated and identified from the medicinal tree *Ginkgo biloba*. Both the fungus and the tree produce bilobalide (**140**) [177,197]. Baker et al. [197] described a concise asymmetric synthesis of (-)-bilobalide; however, the production pathway in the endophytic fungus *P. uvicola* is unknown. Bilobalide (**140**) from the leaves of *Ginkgo biloba* exhibits various functions [198]. For example, as an antioxidant, bilobalide (**140**) affects cerebral ischemia injury by activating the Akt/Nrf2 pathway [199]. Furthermore, Bilobalide (**140**) alleviated morphine-induced addiction in hippocampal neuron cells through the up-regulation of microRNA-101 [200]. In addition, bilobalide (**140**) inhibited 3T3-L1 preadipocyte differentiation and intracellular lipid accumulation [176] and protects ischemia/reperfusion-induced oxidative stress and inflammatory responses via the MAPK/NF-B pathways [201].



### 3.16. Pestalotiopsis vaccinii

*P. vaccinii* (CGMCC3.9199) was isolated from the branches of the mangrove tree *Kandelia candel,* and ten new salicyloid derivatives, namely vaccinols J–S (**141**–**150**), along with five known compounds (*trans*-sordarial, *trans*-sordariol, *cis*-sordariol, 4-hydroxyphthalide, pestalotiopin A), were isolated and identified [32]. All ten compounds were analyzed for their anti-enterovirus 7l (EV71) and cytotoxic activities. Vaccinol J (**141**) exhibited an in vitro anti-EV71 with an IC_50_ value of 30.7 μM. However, none of these new compounds (**141**–**150**) showed cytotoxic effects on the tested cancer cell lines with IC_50_ > 50 μM, including K562 (human erythroleukemic cell line), MCF-7 (human breast cancer cell line), and SGC7901 (human gastric cancer cell line).

### 3.17. Pestalotiopsis versicolor

Two new compounds, a new coumarin, 4,6-dihydroxy-7-formyl-3-methyl coumarin (**151**), and an α-pyrone derivative, 6-[(7*S*,8*R*)-8-propyloxiran-1-yl]-4-methoxy-pyran-2-one (**152**), were found from the plant endophytic fungus *P. versicolor* [179]. In addition, two known compounds, LL-P880g (**153**) and 6-pentyl-4-methoxy-pyran-2-one (**154**), were also found, which were not introduced by Yang XL et al. [21] and Xu J et al. [10]. Bioactivity assay showed that the four compounds (**151**–**154**) did not exhibit significant antifungal activities against the three fungal species, *Fusarium solani*, *Ustilago maydis*, and *C. albicans* [179].



### 3.18. Pestalotiopsis zonata

Two new polyketides, pestalrones A (**155**) and B (**156**), and pestazonatic acid (**157**), were found in the fungus *P. zonata* (CGMCC 3.9222). However, their bioactivities were not analyzed [184].

### 3.19. Pestalotiopsis sp.

Many *Pestalotiopsis* strains were isolated; however, they were not identified carefully, and some bioactive compounds were found in these strains.



From Chinese mangrove *Rhizophora mucronata*, an endophytic *Pestalotiopsis* sp. was isolated, and 11 known compounds were identified, which were not introduced by Yang XL et al. [21] and Xu J et al. [10], including demethylincisterol A_3_ (**158**), dankasterone B (**159**), (22*E*, 24*R*)-ergosta-7,9(11), 22-triene-3β, 5α, 6α-triol (**160**), ergosta-5,7,22-trien-3-ol (**161**), 5, 8-epidioxy-5, 8-ergosta-6, 22E-dien-3-ol (**162**), stigmastan-3-one (**163**), stigmast-4-en-3-one (**164**), stigmast-4-en-6-ol-3-one (**165**), flufuran (**166**), and similanpyrone B (**167**) [140]. Demethylincisterol A_3_ (**158**), ergosta-5,7,22-trien-3-ol (**161**), stigmastan-3-one (**163**), stigmast-4-en-3-one (**164**), stigmast-4-en-6-ol-3-one (**165**), and flufuran (**166**) exhibited significant in vitro cytotoxicity against the human cancer cell lines HeLa, A549, and HepG [140]. Among them, demethylincisterol A_3_ (**158**) was the most potential with IC_50_ values of 0.17, 11.14, and 14.16 nM for the three cell lines, respectively. Ergosta-5,7,22-trien-3-ol (**161**) also showed significant cytotoxicity against HeLa and A549 cell lines with IC_50_ of 21.06 and 11.44 nM, respectively. Flow cytometric investigation also showed that demethylincisterol A_3_ (**158**) mainly inhibited cell cycle at G_0_/G_1_ phase in a dose-dependent manner with significant induction of apoptosis on the human cancer cell lines HeLa, A549, and HepG.

Three new sesquiterpenoids, pestaloporonins A-C (**168**–**170**), related to the caryophyllene-derived punctaporonins, and the known caryophyllene-type metabolite fuscoatrol A (**171**) not introduced by Yang XL et al. [21] and Xu J et al. [10], were isolated from cultures of a fungicolous strain of *Pestalotiopsis* sp., and pestaloporonins A-C (**168**–**170**) did not exhibit effects on *S. aureus*, *B. subtilis*, *B. subtilis*, and *C. albicans* at 50 μg/disk [141]. Fuscoatrol A (**171**) was also found in the fungus *Humicola fuscoatra* and exhibited antimicrobial activities against *S. aureus* and *B. subtilis* (MIC = 12.5 µg/mL) and cytotoxic activity on the developing eggs of the sea urchin *Strongylocentrotus intermedius* (MIC_50_ = 40 µg/mL) [202].



Five new ambuic acid derivatives (**172**–**176**) were found from an endolichenic fungus *Pestalotiopsis* sp. [142]. The suggested biosynthetic pathways of ambuic acid and its derivatives are shown in Figure 5.



The bioactivities of these ambuic acid derivatives (**172**–**176**) were evaluated, and the results showed these compounds (**172**–**176**) did not exhibit significant cytotoxicity against several tumor cells, including A549, HepG 2, and HeLa cell lines (IC_50_ > 40 μM), nor did they show antibacterial activities against *S. aureus*, *B. subtilis*, and *B. subtilis* (MIC > 64 μg/mL) [142]. The antifungal assay showed that compounds **172** and **176** exhibited significant biological effects against *Fusarium oxysporum* with a MIC value of 8 μg/mL. In contrast, compound **176** potently inhibited *Fusarium gramineum* at 8 μg/mL, compared with the positive control ketoconazole (MIC value of 8 μg/mL). These compounds (**172**–**176)** did not exhibit bioactivities against *B. cinerea*, *Alternaria solani*, and *Rhizoctonia solani* (>64 μg/mL).



From the fresh stem bark of *Melia azedarach*, an endophytic fungus, *Pestalotiopsis* sp., was isolated, and eight new caryophyllene sesquiterpenoids named pestaloporinates A-G (**177**–**183**) and 14-acetylhumulane (**184**) were identified from the solid cultures of the fungal strain [143]. These isolated compounds (**177**–**184**) were used to test for their inhibitory effects on nitric oxide production induced by lipopolysaccharide in the murine macrophage RAW 264.7 cell line. The results showed that pestaloporinate B (**178**) exhibited a potent inhibitory effect with an IC_50_ value of 19.0 μM, compared to the positive control (IC_50_ = 40.5 μM), while other compounds were inactive under 50 μM [143].



The endophytic fungus *Pestalotiopsis* sp. was obtained from the leaves of *Photinia frasery* in China. Five new isocoumarin derivatives, pestalactone A–C (**185**–**187**) and pestapyrone D–E (**188**–**189**), together with two known compounds (**190**–**191**), were isolated from the solid cultures of the fungal strain [144]. Their bioactivity assay showed that only pestalactone C (**187**) exhibited potent antifungal activity against *Candida glabrata* (ATCC 90030) with a MIC_50_ value of 3.49 μg/mL. They were inactive against Gram-positive bacteria (*S. aureus* ATCC 25923 and *B. subtilis* ATCC 6633) and Gram-negative bacteria (*B. subtilis* ATCC 25922 and *P. aeruginosa* ATCC 9027) [144].



A *Pestalotiopsis* sp. strain was found from a soft coral, and a pair of new enantiomeric alkaloid dimers, (−)- and (+)-pestaloxazine A (**192**–**193**), were isolated from the fungal strain [147]. (±)- pestaloxazine A, (−)-pestaloxazine A (**192**), and (+)-pestaloxazine A (**193**) showed different antiviral activity against EV71 (enterovirus 71) with IC_50_ values of 16.0, 69.1, and 14.2 μM, respectively. Their selectivity indices (SI) of anti-EV71 activity were 7.9, 2.1, and 9.2, respectively. They did not exhibit any bioactivity against the respiratory syncytial virus (RSV), coxsackie B3 virus (Cox-B 3), and H1N1 virus [147].



Five alkenyl phenol and benzaldehyde derivatives, pestalols A-E (**194**–**198**), were isolated from the endophytic fungus *Pestalotiopsis* sp. AcBC2 deriving from the Chinese mangrove plant *Aegiceras corniculatum* (Myrsinaceae family) [151]. Bioactivity assay of the five compounds showed that (1) pestalol B (**195**) and pestalol C (**196**) exhibited antiproliferative effects at a range of 23.4–42.5 μM against 10 human tumor cell lines, including MCF-7, BT474, A549, DU145, H1975, SK-BR-3, K562, MOLT-4, U937, and BGC823; (2) pestalol A-D (**194**–**197**) showed significant effects on the influenza viruses H3N2 and H1N1 with IC_50_ values of 18.9–48.0 μM [151].



Four novel polyketides, pestalpolyols A-D (**199**–**202**), were also isolated from solid fermentation products of *Pestalotiopsis* sp. cr013 [153]. The skeleton of **202** was almost the same as compound **201**, except for one more carbon in a CH3CH2C = O group. These four compounds (**199**–**202**) did not exhibit any anti-fungal activities against *Gaeumannomyces graminis*, *Fusarium moniliforme*, *Verticillium cinnabarium*, and *Phyricularia oryzae*, nor any anti-bacterial activity against *Pseudomonas solanacearum*, *S. aureus*, and *Salmonella typhimurium* at 100 µg/disk. Neither of them (**199**–**202**) showed any nematicidal activities against *Panagrellus redivivus* and *Caenorhabditis elegans* [153]. Pestalpolyol A (**199**) possessed cytotoxicity against tumor cell lines HL-60, SMMC-7721, A-549, MCF-7, and SW480, with IC_50_ values of 10.4, 11.3, 2.3, 13.7, and 12.4 µM, respectively. Pestalpolyol B (**200**) showed an IC_50_ value of 10.6 µM against A-549, and Pestalpolyol D (**202**) exhibited IC_50_ values of 15.7 µM (HL-60), 31.2 µM (SMMC-7721), 10.7 µM (A-549), 23.7 µM (MCF-7), and 21.4 µM (SW480), respectively [153].



The fungal strain *Pestalotiopsis* sp. cr014 is a mycoparasite of *Cronartium ribicola,* white pine blister rust of *Pinus armandii* in Sichuan Province, China. Nine new polyketides, pestalotic acids A-I (**203**–**211**), were isolated from solid fermentation products of *Pestalotiopsis* sp. cr014 [154]. In addition, pestalotic acid B (**204**), pestalotic acid C (**205**), pestalotic acid G (**209**), and pestalotic acid H (**210**) possessed antibacterial activities with MIC values of 0.78–12.5 µg/mL against *Ralstonia solanacearum* and *Salmonella typhi* [154].



*Pestalotiopsis* sp. PG52 is a mycoparasite isolated from aeciospore piles of *Aecidium pourthiaea,* and four novel polyketides, pestalpolyols E–H (**212**–**215**), were isolated from solid fermentations of this fungal strain [155]. Bioactivity assays showed pestalpolyol F (**213**) possessed cytotoxicity against lung adenocarcinoma cell line A-549 with an IC_50_ value of 11.45 µM; pestalpolyol (**214**) exhibited weak cytotoxicity against four cell lines with IC_50_ values of 14.60 µM (leukemia cell line HL-60), 27.46 µM (hepatocarcinoma cell line SMMC-7721), 11.83 µM (A-549), and 18.50 µM (breast cancer cell line MCF-7); and pestalpolyol (**215**) showed cytotoxicity against three cell line with IC50 values of 22.85 µM (HL-60), 8.05 µM (A-549), and 38.89 µM (MCF-7) [155].

The endophytic fungus *Pestalotiopsis* sp. EJC07 was isolated from *Bauhinia guianensis*, a topical plant of the Amazon. Eight compounds, (4*S*)-4,8-dihydroxy-1-tetralone (**216**), uracil, uridine, *p*-hydroxybenzoic acid, ergosterol, ergosterol peroxide, cerevisterol, and ducitol, were isolated from the fungal strain, with (4*S*)-4,8-dihydroxy-1-tetralone (**216**), which was first reported in the *Pestalotiopsis* genus [203]. This study reported no assay on the bioactivity of (4*S*)-4,8-dihydroxy-1-tetralone (**216**).



From the cultured broth of *Pestalotiopsis* sp. FT172, five ambuic acid derivatives, pestallic acids A–E (**217**–**221**), and (+)-ambuic acid (**222**), were isolated and identified [156]. All the compounds (**217**–**222**) were tested against human cancer cell lines, cisplatin sensitive, and resistant human ovarian carcinoma (A2780S and A2780cisR). Pestallic acid E (**221**) showed potential cytotoxicity with IC_50_ values of 3.3 and 5.1 μM for A2780S and A2780cisR, respectively; and (+)-ambuic acid (**222**) also exhibited inhibition on the two cancer cell lines with IC_50_ values of 10.1 and 17.0 μM, respectively [156].



Cuautepestalorin (**226**) and its putative biosynthetic precursors, cytosporin M (**223**), cytosporin N (**224**), and oxopestalochromane (**225**), were isolated from the bioactive extract of *Pestalotiopsis* sp. [159]. The bioactivity assay showed that oxopestalochromane (**225**) and cuautepestalorin (**226**) showed modest inhibitory activities against α-glucosidase from *S. cerevisiae*, with IC_50_ values of 263.0 and 42.4 μM, respectively, 2 and 14 times more potent than acarbose (604.4 μM), which was used as the positive control [159].



From the solid cultures of the endophytic fungus *Pestalotiopsis* sp. M-23, three new drimane sesquiterpenoids (**227**–**229**), the known 2α-hydroxyisodrimeninol (**230**), and a new isochromone derivative (**231**) were isolated and identified. The bioactivity assay showed that 11-dehydro-3a-hydroxyisodrimeninol (**229**) exhibited a weak inhibitory effect on *B. subtilis* with an IC_50_ value of 280.27 μM and that none of these compounds (**227**–**231**) showed obvious biological activity against *S. aureus* and *M. luteus*. However, drimane sesquiterpenoids were found in other fungi, animals, and plants and showed extensive bioactivities, such as antioxidant, anti-inflammatory, antibacterial, and antifungal [204,205,206,207,208,209,210,211]. Thus, drimane sesquiterpenoids from *Pestalotiopsis* species should be investigated in detail for their bioactivities.

*Pestalotiopsis* sp. Z233 was isolated from the algae *Sargassum horneri*, and two new sesquiterpenes, 1β,5α,6α,14-tetraacetoxy-9α-benzoyloxy-7β*H*-eudesman-2β,11-diol (**232**) and 4α,5α-diacetoxy-9α-benzoyloxy-7β*H*-eudesman-1β,2β,11,14-tetraol (**233**), were isolated from the cultured mycelia of the fungal strain under heavy metal stress (CuCl_2_) [162]. 1β,5α,6α,14-tetraacetoxy-9α-benzoyloxy-7β*H*-eudesman-2β,11-diol (**232**) and 4α,5α-diaacetoxy-9α-benzoyloxy-7β*H*-eudesman-1β,2β,11,14-tetraol (**233**) showed tyrosinase inhibitory activities with IC50 values of 14.8 µM and 22.3 µM (the standard tyrosinase inhibitor kojic acid with IC50 = 21.2 µM), respectively.



Cytosporones were not introduced by Yang XL et al. [21] and Xu J et al. [10]. The endophytic fungus *Pestalotiopsis* sp. was isolated from the leaves of the Chinese mangrove *R. mucronata*, and 11 compounds were isolated, including six cytosporones (**234**–**239**) [146,150]. When these compounds (**234**–**239**) were tested at an initial concentration of 10 µg/mL, none of the compounds showed any significant biological activity against three cancer cell lines, L5178 Y, HeLa, and PC12 [146]. Thus, their bioactivities against bacteria and fungi should be tested.

Endophytic *Pestalotiopsis* sp. BC55 produces exopolysaccharide (EPS), with a maximum EPS value of 4.320 g/L in a 250 ml Erlenmeyer flask containing 75 mL potato dextrose broth supplemented with 7.66 g%/L glucose, 0.29 g%/L urea, and 0.05 g%/L CaCl_2_ with medium pH 6.93, after 3.76 days of incubation at 24 °C [152]. The EPS is a homopolysaccharide of (1 → 3)-linked-d-glucose. EPSs are also produced by other fungi and bacteria, such as *F. solani* [212], *F. oxysporum* [213], *Stemphylium* sp. [214], the mangrove endophytic fungus *Aspergillus* sp. Y16 [215], lactic acid bacteria [216,217], *Bacillus mycoides* [218], and *Bacillus licheniformis* [219]. EPS bioactivity greatly varied due to chain length, molecular weight, branching, etc. A bioactive EPS with Mw ~1.87 × 105 Da was isolated from endophytic fungus *F. solani* SD5 [212]. The isolated EPS showed in vitro anti-inflammatory and anti-allergic activity, and the EPS (1000 μg/mL) protects 55% of erythrocytes from hypotonic solution-induced membrane lysis. EPS produced by *B. mycoides* exhibited an anti-tumor effect [218]. EPS produced by *B. mycoides* showed low cytotoxicity against normal cells of baby hamster kidney (BHK) with an IC_50_ value of 254 μg/mL, while it exhibited an inhibitory effect against cancer cells of human hepatocellular carcinoma (HepG2) and colorectal adenocarcinoma cells (Caco-2) with IC50 of 138 μg/mL and 159 μg/mL, respectively. Ren Q et al. [220] purified EPSs with a molecular weight of 2.7 × 10^6^ Da to 1.7 × 10^7^ Da from *Lactobacillus casei* and found that EPSs promote the differentiation of CD4 T lymphocytes into T-helper 17 cells in BALB/c mouse Peyer’s patches in vivo and in vitro. Thus, it is reasonable to speculate that EPSs produced by *Pestalotiopsis* species might exhibit various similar bioactivities.

## 4. Accurate Biosynthesis Pathways and Enhanced Accumulation of Secondary Metabolites in *Pestalotiopsis*

Nutritional and environmental factors greatly promote secondary metabolite biosynthesis in *Pestalotiopsis* species [80,83,85]. Under the best nutritional and environmental conditions, how to maximize the yield of secondary metabolites in *Pestalotiopsis* species is a key problem. Genetic modification in biosynthesis pathways of important secondary metabolites is the best choice with certainty. The aim of genetic modification is to increase or inhibit the activities of key enzymes in biosynthesis pathways of wanted secondary metabolites in order to increase their yield. Some key enzymes in the biosynthesis of secondary metabolites in *Pestalotiopsis* species have been identified to date, improving our knowledge about accurate biosynthesis pathways and enhancing the accumulation of novel secondary metabolites.

### 4.1. Transcription Factors Involved in Secondary Metabolite Biosynthesis in Pestalotiopsis

Given the roles of transcription factors in gene expression, transcription factors involved in the biosynthesis of secondary metabolites in *Pestalotiopsis* species have been widely studied. Two transcription factors, *PfmaF* and *PfmaH*, cooperatively regulate 1,8-dihydroxy naphthalene (DHN) melanin biosynthesis in *P. fici*. *PfmaH*, as a pathway-specific regulator, mainly regulates melanin biosynthesis, and PfmaF functions as a broad regulator to stimulate *PfmaH* expression in melanin production [90]. In addition, PfmaH directly regulates the expression of scytalone dehydratase, which catalyzes the transition of scytalone to 1,3,8-trihydroxynaphthalene (T3HN), which is reduced to vermelone, and vermelone is converted into DHN. Zhang P et al. [90] disrupted the gene *PfmaF* using the CRISPR/Cas9 system. They found that the disruption affected neither DHN melanin distribution nor conidia cell wall integrity in *P. fici*. Yet, the overexpression of *PfmaF* leads to heavy pigment accumulation in *P. fici* hyphae. Recently, two new transcription factors, Pmr1 and Pmr2, were identified in *P. micropspora* [221]. *Pmr1* and *Pmr2* were located in the gene cluster for melanin biosynthesis and both of them regulated the expression of genes in the melanin biosynthesis cluster. In *Δpmr1* and *Δpmr2* mutant strains, most genes in the gene cluster (including 21 genes, i.e., GEM11355_g–GEM11375_g) were significantly upregulated. Their upregulation is related to increased yield of secondary metabolites in the mutant strains *Δpmr1,* compared with the wild type (WT). Meanwhile, HPLC analysis showed that the pestalotiollide B peak at 3.3 min was much greater in the *Δpmr1* and *Δpmr2* strains than that in WT; moreover, this increment in *Δpmr1* was significantly greater than that in *Δpmr2*. In addition, Pmr1 played a larger regulatory role in secondary metabolism than Pmr2.

PfZipA, on the other hand, is one of the bZIP transcription factors in *P. fici.* Without oxidative treatment, the *ΔPfzipA* mutant strain of *P. fici* produced less isosulochrin and ficipyrone A than wild type [78]. However, PfZipA mediates the sensitivity of *P. fici* to oxidative stress caused by the oxidative reagents that-butyl hydroperoxide (tBOOH), diamide, H_2_O_2_, and menadione sodium bisulfite (MSB). tBOOH treatment decreased the production of iso-A82775C and pestaloficiol M in *ΔPfzipA* strain; MSB treatment decreased the production of RES1214-1 and iso-A82775C; however, it increased pestaloficiol M production in the mutant; and H_2_O_2_ treatment resulted in enhanced production of isosulochrin, RES1214-1, and pestheic acid (**23**), yet decreased ficipyrone A and pestaloficiol M in *ΔPfzipA* strain, compared to the wild type [78].

### 4.2. Histone Acetylation

Histone acetylation is an important modification of histone proteins, which plays an important role in condensing and relaxing DNA. Histone acetylation is also involved in the biosynthesis of secondary metabolites in *Pestalotiopsis* species. Zhang Q et al. [113] identified a B-type histone acetyltransferase, Hat1, in the *P. microspora*. Secondary metabolites dramatically decreased in a *hat1* deletion mutant strain, suggesting HAT1 functions as a regulator of secondary metabolism. Therefore, it is reasonable to speculate that the overexpression of the gene *hat1* improves the biosynthesis of secondary metabolites in the fungus, thus, its overexpression mutant strains might be used for specific metabolites. In *P. microspora*, an MYST histone acetyltransferase encoded by the gene *MST2* modulates secondary metabolism and conidial development [222]. Deleting the gene (*mst2*) caused serious growth retardation and impaired conidial development, e.g., delayed and reduced conidiation and aberrant conidia capacity. At the same time, overexpression of *mst2* triggered earlier conidiation and higher conidial production. Deletion of *mst2* also reduced the production of secondary metabolites in *P. microspora* [222]. In *P. microspora* NK17, Niu X et al. [112] found that a putative histone deacetylase gene (*HID1*) played an important role in the biosynthesis of pestalotiollide B. In the *hid1* null mutant, the yield of pestalotiollide B increased approximately 2-fold to 15.90 mg/L. In contrast, the deletion of gene *hid1* resulted in a dramatic decrease in conidia production of *P. microspora* NK17. These results suggest that the histone deacetylase *HID1* is a regulator, concerting secondary metabolism and development, such as conidiation, in *P. microspora.*

### 4.3. Polyketide Synthases

Polyketides possess diverse chemical structures and biological activities and are the most important sources of novel secondary metabolites in plants, bacteria, and fungi. Polyketide synthases (PKSs) catalyze the biosynthesis of polyketides. While type I and type II PKSs exist as large protein complexes, type III PKSs are relatively small homodimeric proteins (~45 kDa monomer). In *Pestalotiopsis* species, PKSs are involved in the biosynthesis of some secondary metabolites. For example, the biosynthesis of pestalotiollide B is controlled by polyketide synthase [111]. Chen L and co-workers successfully deleted 41 out of 48 putative PKSs in the genome of *P. microspora* NK17. Furthermore, they found that 9 of the 41 PKS deleted strains significantly increased the biosynthesis of pestalotiollide B, and the deletion of *pks35* increased pestalotiollide B by 887% [111].

The fungal products dibenzodioxocinones promise a novel class of inhibitors against cholesterol ester transfer protein [112]. A gene cluster of 21 genes, including *PKS8* encoding a polyketide synthase, was defined, and disruption of genes in the cluster led to the biosynthesis of loss of dibenzodioxocinones [120]. Of the 21 genes, 5 genes, i.e., *GME11356*, *GME11357*, *GME11358*, *GME11365*, and *GME11367*, were deduced to participate in the generation of the backbone structure, and three regulatory genes, i.e., *GME11360*, *GME11369*, and *GME11370*, were also identified.

After forming polyketides, they can be converted into other secondary metabolites. The pestheic acid biosynthetic gene (*pta*) cluster was identified through genome scanning of the fungus *P. fici*. The biosynthetic pathway was elucidated through gene disruption intermediate detection and enzymatic analysis [82]. The results showed that the pestheic acid biosynthesis proceeded through the formation of the polyketide backbone, cyclization of a polyketo acid to a benzophenone, chlorination, and construction of the diphenyl ether skeleton through oxidation and hydrolyzation. The gene *PTAA* is important in pestheic acid biosynthesis in *P. fici*. Pestheic acid was abolished in the *ptaA* disruption mutants of *P. fici* [82]. In the pestheic acid biosynthesis pathway, the gene *PTAM* encodes a flavin-dependent halogenase, catalyzing chlorination. Inactivation of flavin-dependent halogenase from the *Chaetomium chiversii* radicicol locus yielded dechloro-radicicol (monocillin I) [223]. Thus, in *P. fici*, *PTAM* (*ptaM*) disruption might result in a change in pestheic acid biosynthesis.

### 4.4. Other Regulatory Proteins and Enzymes

The Snf1/AMPK is highly conserved in the eukaryotes and acts as a central regulator of carbon metabolism and energy production. In the filamentous fungus *P. microspora*, *SNF1* concerts carbon metabolism and filamentous growth, conidiation, cell wall integrity, stress tolerance, and the biosynthesis of secondary metabolites [224]. The *Snf1* deletion strain of *P. microspora* NK17 (*∆snf1*) displayed remarkable retardation in vegetative growth and pigmentation. Furthermore, it produced a diminished number of conidia, even in the presence of glucose, and *Snf1* deletion caused damage to the cell wall of *P. microspora* [224]. In addition, Pestalotiollide B was considerably reduced in the mutant strain *∆snf1.* These results demonstrate that SNF1 is a regulator of secondary metabolism and may be involved in either the activation or silencing of certain gene clusters in *P. microspora* NK17. Therefore, the more accurate function of *SNF1* should be elucidated in secondary metabolite biosynthesis research.

Evidence shows biosynthesis of secondary metabolites and development are correlated processes in fungi, and pleiotropic proteins regulate the equilibrium between the biosynthesis of secondary metabolites and development. A global regulator, *RsdA* (regulation of secondary metabolism and development), was identified through genome-wide analysis and deletion of the regulator gene in the endophytic fungus *P. fici* [225]. Deleting *rsdA* significantly reduced asexual development, resulting in low sporulation, abnormal conidia, and major secondary metabolites (such as asperpentyn, fificiolide A, and chloroisosulochrin) while remarkably increasing melanin pigment production. In addition, pestheic acid, a basic building block for a group of structurally diverse compounds, was completely abolished in the *ΔrsdA* strain, implying that the biosynthesis of pestheic acid analogs was dramatically reduced.

Canonical Gcn2/Cpc1 kinase is an amino acid sensor and regulates the expression of target genes in response to amino acid starvation. When the mutant strain *Δgcn2* of *P. microspora* was cultured in the presence of 3AT (5 mM) to mimic amino acid starvation conditions, biosynthesis of pestalotiollide B was almost inhibited [114]. Meanwhile, the loss of *gcn2* led to a less-pigmented phenotype of *P. microspora* [114]. All the results demonstrate that the protein encoded by *gcn2* is a regulator of secondary metabolism and may be involved in either activation or silencing of gene clusters in *P. microspora*.

G-protein-mediated signaling pathways regulate fungal morphogenesis, development, pathogenesis, and secondary metabolism [226,227,228,229,230,231]. In *Pestalotiopsis* species, G protein-mediated signaling regulates secondary metabolites. The gene *pgα1* putatively encodes the α-subunit of a group I G protein in *P. microspora* NK17. The *pgα1* deletion mutants showed retarded vegetative growth, mycelium aging, premature conidiation, deformed conidia, significantly increased melanin production, and a sharp decrease in the production of pestalotiollide B [13]. Meanwhile, the expression of *pks1*, which encodes melanin polyketide synthase involved in 1,8-dihydroxy naphthalene (DHN) melanin biosynthesis, was upregulated 55-fold in *pgα1* deletion mutants. All the results imply obvious changes in the biosynthesis of different secondary metabolites in *pgα1* mutants. In addition, the deficiencies of pestalotiollide B production and conidiation in *Δpgα1* mutants could not be rescued by deletion or overexpression of the gene *hid1* encoding histone deacetylase, suggesting that the protein PGα1 can override the effect of *hid1* on pestalotiollide B production and conidiation.

In the fungus *P. microspora*, two genes, *choA* and *choC,* encode two phospholipid methyltransferases. *choC* deletion mutants (*choC^Δ^*) resulted in defects in phosphatidylcholine production, vegetative growth, and development of asexual structure [49], suggesting that genetic modification might regulate secondary metabolite biosynthesis in *Pestalotiopsis* species. However, *choA*, but not *choC*, was required to produce pestalotiollide B [49], suggesting distinct roles of the two genes.

The earlier examples demonstrate changes in the biosynthesis of secondary metabolites in *Pestalotiopsis* species by molecular tools, especially gene editing. Therefore, key genes encoding important enzymes in secondary metabolite biosynthesis in *Pestalotiopsis* species should be cloned, and the overexpression or deletion of these key genes is useful for enhanced biosynthesis of important secondary metabolites. More importantly, accurate biosynthesis pathways of secondary metabolites are the premise. Based on these basic studies on the effects of secondary metabolites in *Pestalotiopsis* species on human health, animals, and plants and the identification of their accurate biosynthesis pathways, it is possible to enhance biosynthesis and the accumulation of key secondary metabolites in the future. The industrial production of important secondary metabolites in this way will become possible.

## 5. Concluding Remarks and Future Perspectives

Given the important effects of secondary metabolites from *Pestalotiopsis* species on human health, animals, and plants, two aspects, i.e., the effects of these secondary metabolites and their accurate biosynthesis pathways, are vital. Therefore, more studies should focus on their accurate biosynthesis pathways to enhance biosynthesis and accumulation, further establishing the foundation for the industrial production of secondary metabolites from *Pestalotiopsis* species. Gene editing is a valuable method for fully comprehending secondary metabolite biosynthesis processes; however, it is very difficult to establish gene-editing systems for some *Pestalotiopsis* species, despite genome editing systems having been established for few *Pestalotiopsis* species, such as *P. fici* and *P. microspora* [90,232,233,234]. Furthermore, more effective gene-editing tools are to be developed and, therefore, long-term efforts are in the pipeline.

In addition, improvements for the best growth conditions are useful for enhanced biosynthesis and accumulation of secondary metabolites. For example, the addition of some chemicals in the culture medium promotes the biosynthesis of secondary metabolites, such as salicylic acid [235]. Meanwhile, the co-cultivation of fungi and bacteria can also trigger the biosynthesis of secondary metabolites. For example, the co-cultivation of *Aspergillus flavipes* and *B. subtilis* triggers the biosynthesis machinery of taxol [236]. At present, are isa no reports on the co-cultivation of *Pestalotiopsis* species with other microbes. Many gene clusters for the biosynthesis of secondary metabolites in filamentous fungi often stay silent under some culture conditions because of the absence of interaction with bacteria. For instance, Brakhage and colleagues have discovered that the silent secondary metabolite gene cluster for orsellinic acid (*ors*) in the filamentous fungus A*spergillus nidulans* is activated upon physical interaction with the bacterium *Streptomyces rapamycinicus,* and the interaction of the fungus with this distinct bacterium led to increased acetylation of histone H3 lysines 9 and 14 at the *ors* gene cluster, thus to its activation [237,238,239]. Then, they identified the Myb-like transcription factor BasR, a master regulator of bacteria-triggered production of fungal secondary metabolites, by chromatin mapping [240]. However, the interaction between *Pestalotiopsis* species and bacteria and key regulator nodes for transduction of the bacterial signals in the fungi is unclear. Certainly, activating silent gene clusters in *Pestalotiopsis* species is a good strategy for enhanced biosynthesis and accumulation of fungal secondary metabolites, just as in the Brakhage and Schroeckh advocated strategies [241]. Furthermore, as mentioned above, gene editing is a good and useful approach to increase the yield of secondary metabolites. We should try our utmost to establish whole feasible systems of gene editing for important *Pestalotiopsis* species. At present, the *Pestalotiopsis* species investigated are only a small part of this genus, and more species are yet to be studied and developed for human health.

## Figures and Tables

**Figure 1 molecules-27-08088-f001:**
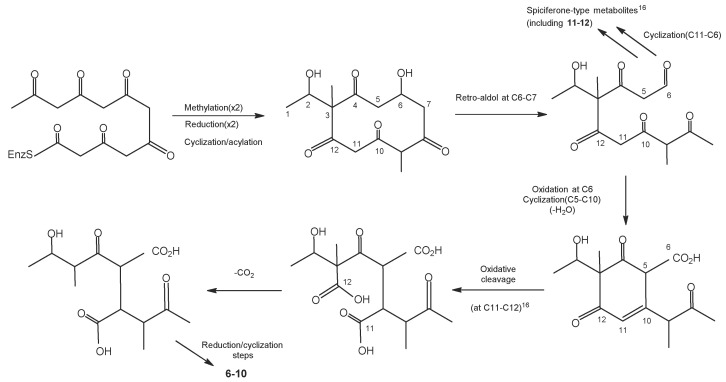
Synthesis pathway of disseminins and spiciferones.

**Figure 2 molecules-27-08088-f002:**
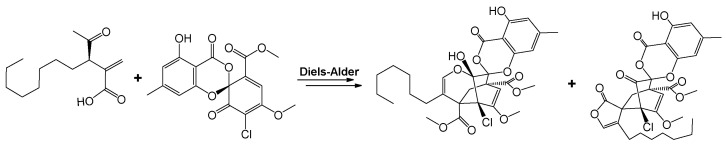
Conversion of 1-undecene-2,3-dicarboxylic acid and maldoxin to chlorotheolides via Diels-Alder reactions [170].

**Figure 3 molecules-27-08088-f003:**
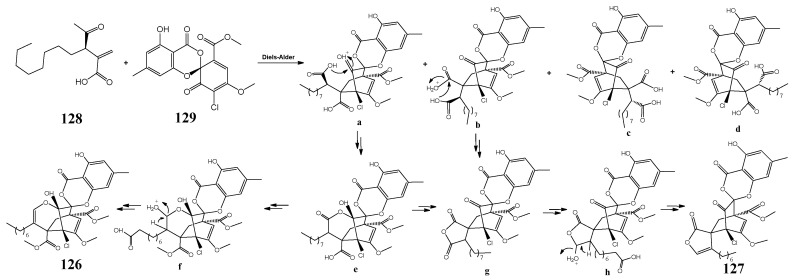
The hypothetical biosynthetic pathways for chlorotheolides [170].

**Figure 4 molecules-27-08088-f004:**
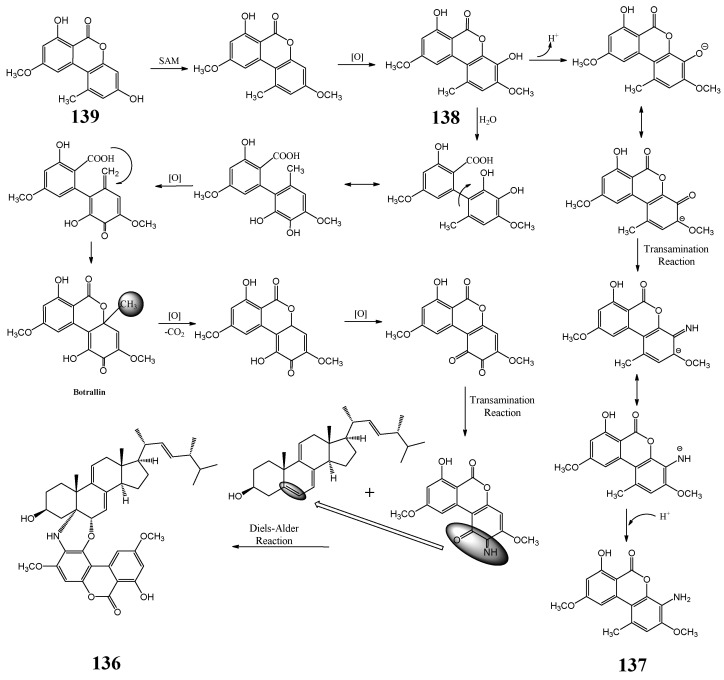
Plausible biogenetic correlation of pestauvicomorpholine A and three alternariol analogues [175].

**Figure 5 molecules-27-08088-f005:**
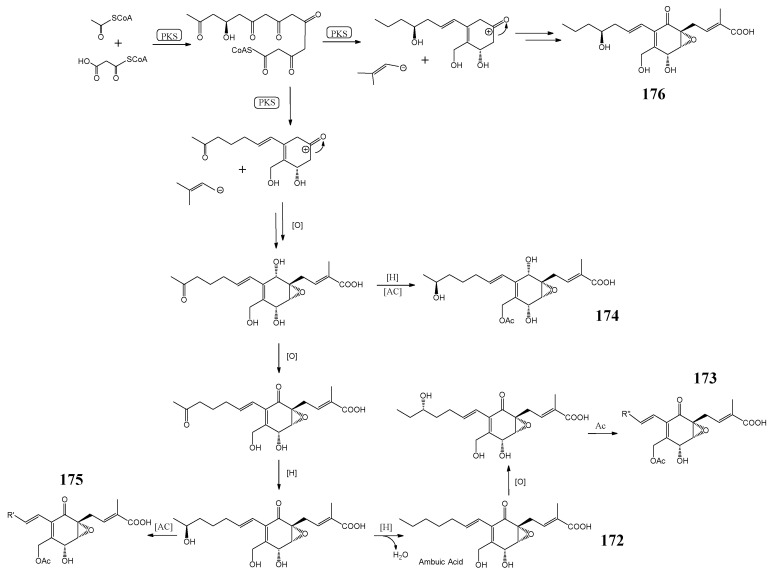
The suggested biosynthetic pathways of ambuic acid and its derivatives [142].

**Table 1 molecules-27-08088-t001:** Identified metabolites from *Pestalotiopsis* species and their bioactivities.

Fungal Species	Metabolites	Bioactivity	References
*P. adusta*	pestalachlorides A–C	antifungal	[69]
	diterpenoid		[70]
*P. besseyi*	furanones		[71]
*P. breviseta*	taxol	anticancer	[16,17]
*P. crassiuscula*	a new coumarin and six known compounds		[72]
*P. diploclisia*	pestalotioquinols G and H, pestalotioquinol A, phomonitroester, (*R*)-4,6,8-trihydroxy-3,4-dihydronaphthalen-1(2*H*)-one, and scylatone	antimalarial and cytotoxic activity	[73]
*P. disseminata*	6-hydroxypunctaporonin E, 6-hydroxypunctaporonin B, and 6-hydroxypunctaporonin A	anti-bacteria	[74]
	disseminins A–E, spiciferones D and E		[75]
*P. fici*	chloropupukeananin and chloropestolides	antimicrobial, antitumor, and anti-HIV activities	[76]
	pestaloficiols A–E	inhibitory effects on HIV-1 replication	[77]
	isosulochrin, ficipyrone A, pestheic acid, iso-A82775C, pestaloficiol M, RES1214-1, and iso-A82775C		[78]
	pestalofones A–E, isosulochrin, isosulochrin dehydrate, and iso-A82775C	inhibitory effects on HIV-1 replication and antifungal activity	[79]
	chloropestolide A	antitumor	[80]
	chloropupukeanolides C–E	cytotoxicity	[81]
	pestheic acid		[82]
	chloropupukeananin		[83]
	melanin		[84]
	chloropupukeanolides A and B, chloropupukeanone A	anti-HIV-1 and cytotoxic activity	[85]
	isoprenylated chromone derivatives		[86]
	Chloropestolides B–G		[87]
	pestaloficins A–E		[88]
	pestaloficiols Q–S		[89]
	DHN melanin		[90]
	2H-pyran-2-one and 2H-furan-2-one derivatives		[91]
*P. flavidula*	spiroketal derivatives	cytotoxicity	[92]
*P. foedan*	pestafolide A, pestaphthalide A and B	antifungal activity	[93]
	(−)-(4*S*, 8*S*)-foedanolide and (+)-(4*R*, 8*R*)-foedanolide		[94]
	monoterpene derivatives	antifungal	[95]
*P. guepinii*	metabolites of ciprofloxacin and norfloxacin		[96]
	culture broth extract	inhibit actinomycete growth	[97]
	pestheic acid or dihidromaldoxin	genotoxicity and mutagenicity	[98]
	alpha-pyrones		[99]
*P. hainanensis*	taxol		[19]
	caryophyllene-Type Sesquiterpenes		[100]
*P. heterocorni*	7-hydroxy-5-methoxy-4,6-dimethyl-7-*O*-α-L-rhamnosyl-phthalide and 7-hydroxy-5-methoxy-4,6-dimethyl-7-*O*-β-D-glucopyranosyl-phthalide		[101]
	heterocornols A–L, methyl-(2-formyl-3-hydroxyphenyl) propanoate, cladoacetal A, xylarinol A, agropyrenol, vaccinol G, (*R*)-3-hydroxy-1-[(*R*)-4-hydroxy-1,3-dihydroisobenzofuran-1-yl]butan-2-one, and (*R*)-3-hydroxy-1-[(*S*)-4-hydroxy-1,3-dihydroisobenzo -furan-1-yl]butan-2-one	cytotoxicity and antifungal	[29]
	pestaloisocoumarins A and B, isopolisin B, pestalotiol A, gamahorin, pestalachloride B, pestalachloride E, pestalalactone atropisomers (8a/8b),	cytotoxicity	[30]
	heterocornols M and N, heterocornols O and P	anticancer	[28]
*P. humus*	pestynol	antibacterial and antifungal activity, weak cytotoxicity	[102]
	pestiocandin	antibacterial and antifungal activity	[103]
*P. jesteri*	jesterone and hydroxy-jesterone	selective antimycotic activity	[104]
*P. karstenii*	pestalrone A, pestalrone B, pestalotin, hydroxypestalotin	cytotoxic activity, antiprotozoal activity	[42]
*P. leucothës*	BS, GS, and YS	Immunomodulatory	[105]
		immunomodulatory effects	[106]
*P. mangiferae*	4-(2,4,7-trioxa-bicyclo [4.1.0] heptane-3-yl) phenol	antibacterial and antifungal activity	[107]
*P. microspora*	taxol	antiproliferative activity	[11,108]
	ɑ-pyrone		[109]
	(+)-dendocarbin L, (+)-sydonic acid, and (+)-sydowic acid	cytotoxicity	[68]
	isopestacin	antifungal and antioxidant activities	[110]
	pestalotiollide B		[49,111,112,113]
	pestalotiollide B, melanin		[114]
	7-epi-10-deacetyltaxol	induces apoptosis	[115]
	taxol, pestalotiollide B, 1, 8-dihydroxy naphthalene melanin	antitumor	[13]
	melanin		[116]
	ursolic acid		[117]
	pestalotioprolides C, D–H, 7-*O*-methylnigrosporolide, pestalotioprolide B, seiricuprolide, nigrosporolide, and 4,7-dihydroxy-13-tetradeca-2,5,8-trienolide	cytotoxic	[118]
	dibenzodioxocinons	inhibitors of cholesterol ester transfer protein	[119,120]
	pestalotioquinols A and B	neuroprotective	[121]
	pitholide E, pitholide B, pitholide D, pestalotin, PC-2, tyrosol, 4-oxo-4*H*-pyran-3-acetic acid		[67]
	2H-pyranone and isocoumarin derivatives	antifungal	[66]
	Microsporols A–C, ambuic acid	5-lipoxygenase (5-LOX) inhibitory effects	[122]
*P. neglecta*	ambuic acid derivatives	inhibitory activity against the NO production	[123]
	crude methanol and ethyl acetate extract	antibacterial activity	[124]
	pestalotiochromenoic acids A–D, pestalotiochromones A and B	liver X receptors modulators	[125]
	ambuic acid	anti-inflammatory action	[126]
	benzophenones	inhibit pancreatic cancer cells	[31]
	Ene-yne Hydroquinones		[26]
	pestalopyrones A–D		[127]
	pestallic acids F and G, pestalotiopyrone N, neopestalone, sesquicaranoic acid B, monocycloalternarene B, pestalone, 2,4-dihydroxy-3,5,6-trimethyl benzoic acid, and citreorosein		[27]
*P. palmarump*	sinopestalotiollides A–D, 3′-*O*-methyldehydroisopenicillide, ∆1′3′-1′-dehydroxypenicillide, dehydroisopenicillide, 2′-hydroxy-3′,4′-didehydropenicillide, scirpyrones A, 5,6-dihydro-4-methoxy-2H-pyran-2-one, LL-P880α, (6*S*,1′*S*,2′*R*)-LL-P880b, photipyrone B, PC-2, (1′*S*,2′*R*)-LL-P880γ, necpyrone C	cytotoxic	[128]
*P. pauciseta*	taxol	anticancer	[15]
*P. photiniae*	phthalide derivatives	against plant pathogens	[129]
	photipyrones A, B, C	modest inhibitory effects on the growth of MDA-MB-231	[130]
	4-(3′,3′-dimethylallyloxy)-5-methyl-6-methoxyphthalide	induced G1 cell cycle arrest and apoptosis in a dose-dependent manner	[131]
	4-(3′,3′-Dimethylallyloxy)-5-methyl-6-methoxy-phthalide	cytotoxicity	[132]
	three new phthalide derivatives and six known phthalide derivatives	antifungal activities	[133]
	photinides A–F	cytotoxic	[52]
*Pestalotiopsis* sp.	RES-1214-1 and -2	non-peptidic endothelin type A receptor antagonists	[134]
*Pestalotiopsis* sp.	pestalotiopamide E		[135]
*Pestalotiopsis* sp.	pestalotiopens A and B		[136]
*Pestalotiopsis* sp.	pestalactams A–C		[137]
*Pestalotiopsis* sp.	pestaloficiol J, (±)-pestaloficiol X		[138]
*Pestalotiopsis* sp.	pestaloquinols A and B		[139]
*Pestalotiopsis* sp.	demethylincisterol A_3_, dankasterone B, (22*E*, 24*R*)-ergosta-7,9, 22-triene-3β, 5α, 6α-triol, ergosta-5,7,22-trien-3-ol, 5,8-epidioxy-5,8-ergosta-6,22*E*-dien-3-ol, stigmastan-3-one, stigmast-4-en-3-one, stigmast-4-en-6-ol-3-one, flufuran, (2-*cis*, 4-*trans*)-abscisic acid, similanpyrone B.	induce G0/G1 cell cycle arrest and apoptosis in human cancer cells	[140]
*Pestalotiopsis* sp.	pestaloporonins		[141]
*Pestalotiopsis* sp.	polyketide-terpene hybrid metabolites		[142]
*Pestalotiopsis* sp.	pestaloporinates A–G and 14-acetylhumulane		[143]
*Pestalotiopsis* sp.	isocoumarin derivatives	antifungal	[144]
*Pestalotiopsis* sp.	pestalotiopsolide A, taedolidol and 6-epitaedolidol		[41]
*Pestalotiopsis* sp.	(±)-pestalachloride D	antibacterial	[145]
*Pestalotiopsis* sp.	cytosporones J-N, pestalasins A-E, pestalotiopsoid A, cyclosporine C, dothiorelone B, and 3-hydroxymethyl-6,8-dimethoxycoumarin (13).		[146]
*Pestalotiopsis* sp.	(+)- and (−)-pestaloxazine A	antiviral	[147]
*Pestalotiopsis* sp.	pestalotiopsins A and B	immunosuppressive	[148]
*Pestalotiopsis* sp.	ambuic acid and torreyanic acid derivatives	antimicrobial activity	[149]
*Pestalotiopsis* sp.	pestalotiopsones A–F	moderate cytotoxicity	[150]
*Pestalotiopsis* sp. AcBC2	pestalols A–E, 4-hydroxyphenethyl 2-(4-hydroxyphenyl) acetate, r-hydroxyphenyl acetic acid methyl ester, transharzialactones A and F, 3-hydroxy-3-methyl-d-lactone, 3β,5α, 9α-trihydroxy-7, 22-en-ergost-6-one, and 3b-hydroxy-sterol	cytotoxicity, inhibitory activities against Influenza A virus subtype (H3N2), Swine Flu (H1N1) viruses, tuberculosis	[151]
*Pestalotiopsis* sp. BC55	exopolysaccharide		[152]
*Pestalotiopsis* sp. cr013	pestalpolyols A–D	cytotoxic	[153]
*Pestalotiopsis* sp. cr014	pestalotic acids A–I	antibacterial	[154]
*Pestalotiopsis* sp. PG52	pestalpolyols E–H	cytotoxic activities	[155]
*Pestalotiopsis* sp. EJC07	(4*S*)-4,8-dihydroxy-1-tetralone, uracil, uridine, *p*-hydroxybenzoic acid, ergosterol, ergosterol peroxide, cerevisterol and ducitol		[74]
*Pestalotiopsis* sp. FT172	pestallic acids A–E	anti-proliferative	[156]
*Pestalotiopsis* sp. HC02	pestalotines A and B		[38]
*Pestalotiopsis* sp. HHL101	pestalotiopisorin B		[157]
*Pestalotiopsis* sp. HQD-6	pestalotiopin B and pestalotiopyrone N	very weak cytotoxic	[158]
*Pestalotiopsis* sp. IQ-011	cuautepestalorin, cytosporin M, cytosporin N, oxopestalochromane, pestalone	inhibitory properties against α-glucosidase from S. cerevisiae	[159]
*Pestalotiopsis* sp. M-23	drimane sesquiterpenoids, 2α-hydroxyisodrimeninol, a new isochromone derivative	weak antibacterial	[160]
*Pestalotiopsis* sp. PSU-MA69	pestalochromones A–C, pestalotethers A–D, pestaloxanthone, pestalolide	antifungal activity against *Candida albicans* and *Cryptococcus neoformans*	[161]
*Pestalotiopsis* sp. Z233	1β,5α,6α,14-tetraacetoxy-9α-benzoyloxy-7β *H*-eudesman-2β,11-diol and 4α,5α-diacetoxy-9α-benzoyloxy-7β*H*-eudesman-1β,2β,11,14-tetraol	tyrosinase inhibitory activities	[162]
*Pestalotiopsis* spp.	ambuic acid	antifungal	[163]
*Pestalotiopsis* spp.	chromones, cytosporones, polyketides, terpenoids and coumarins		[164]
*P. sydowiana*	1-*O*-methyldehydroisopenicillide, pestalotiollide B, pestalotiollide A, dehydroisopenicillide, 6-hydroxymethyl-4-methoxy-5,6-dihydro-2*H*-pyran-2-one, pestalotiopyrone D, pestalotiopyrone E, pestalotiopyrone G, LL-P880b, and photipyrone B	20S proteasome inhibitory activities	[165]
	cyclo(-Leu-Pro) and 4-hydroxyphenylacetamide	antimicrobial,	[33]
*P. terminaliae*	taxol	anticancer	[18]
*P. theae*	chloroisosulochrin, ficipyrone A and pestheic acid	strong activity against respiratory syncytial virus	[166]
	pesthetoxin		[167]
	pestalotheols A–D	inhibitory effect on HIV-1 LAI replication	[168]
	pestalazines and pestalamides	inhibitory effects on HIV-1 replication and antifungal activity	[169]
	chlorotheolides A and B, 1-undecene-2,3-dicarboxylic acid, maldoxin		[170]
	pestalotiones A–D		[171]
	chloropupukeananin and pestalofone C	regulate autophagy through AMPK and Glycolytic Pathway	[172]
	pestathenols A and B, pestatheranone A, punctaporonins O, P, and R, ficifuranone B, and decarestrictine D		[173]
	pestalothenin A–C	cytotoxic and antibacterial activity	[174]
*P. uvicola*	pestauvicomorpholine A and three alternariol analogues	cytotoxicity against mouse melanoma (B16-BL6) cell line	[175]
	bilobalide	suppresses adipogenesis in 3T3-L1 adipocytes via the AMPK signaling pathway; protect BV2 microglia cells against OGD/reoxygenation injury	[176,177,178]
*P. vaccinii*	vaccinols J-S, *trans*-sordarial, *trans*-sordariol, *cis*-sordariol, 4-hydroxyphthalide, pestalotiopin A		[32]
*P. versicolor*	4,6-dihydroxy-7-formyl-3-methyl coumarin, 6-[(7*S*,8*R*)-8-propyloxiran-1-yl]-4-methoxy-pyran-2-one	devoid of significant antifungal activity against *F. solani*, *Ustilago maydis*, and *C. albicans*	[179]
*P. virgatula*	pestalospiranes A and B		[180]
	9-hydroxybenzo[C]oxepin-3[1*H*]-one		[181]
	virgatolides A–C		[182]
*P. yunnanensis*	pestalotic acids A–G	significant antimicrobial activity	[183]
*P. zonata*	pestalrones A and B, pestalotin, hydroxypestalotin, pestazonatic acid, necpyrones A–B		[184]

## Data Availability

Not applicable.

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
