# Peer review of "Pestalotiopsis Diversity: Species, Dispositions, Secondary Metabolites, and Bioactivities"

_molecules, 2022, doi:10.3390/molecules27228088_

Round 1
Reviewer 1 Report
The manuscript collects the results relating to the species, dispositions, secondary metabolites, and bioactivities of Pestalotiopsis. A remarkable work for the amount of data shown, well organized and certainly useful for researchers working in this field. Authors should only check that they have used the same font throughout the manuscript, use abbreviations after first reporting the name of a plant or a test organism, check that the structures are always spelled the same and the names of the compounds are correct. In the attached file are highlighted a few things to correct and among these perhaps to shorten the first part of the introduction.

Author Response
To Editors and Reviewers
I am so sorry for delaying the revised submission, because our city was quarantined for about 10 days, I could not go to my office and deal with the manuscript. Before the quarantine, we had finished most revision, only structures of two molecules left. This morning I went to the campus, and asked my student to carry out my computer and brought it back to my house, because I was still not allowed to come into the campus.
According to opinions from reviewers, we revised the manuscript. In addition, we found some other errors and corrected them.
Below is the responses to Reviewer 1
The manuscript collects the results relating to the species, dispositions, secondary metabolites, and bioactivities of Pestalotiopsis. A remarkable work for the amount of data shown, well organized and certainly useful for researchers working in this field. Authors should only check that they have used the same font throughout the manuscript, use abbreviations after first reporting the name of a plant or a test organism, check that the structures are always spelled the same and the names of the compounds are correct. In the attached file are highlighted a few things to correct and among these perhaps to shorten the first part of the introduction.
- We deleted the part in Introduction highlighted by the reviewer, and added a sentence in stead. And subsequently we deleted the references cited in this part.
- We checked the Latin names of the fungi in Pestalotiopsisgenus and plant species. Their names occurring in subsequent parts of the article were abbreviated.
We checked the names and structures of the metabolites and assured that these names and structures are correct.
Reviewer 2 Report
The authors describe the origin of the compounds isolated from the Pestalotiopsis genus, their biological importance, chemical structures and even some metabolite synthesis. It is an exhaustive and interesting revision, however there are some errors that must be taken care of, and they are mentioned below:
Stereochemical descriptors should be used appropriately (R,S, cis, trans, Z, E), Table 1, and the full manuscript.
Check carefully all the structures and names of the molecules, for example naphthalen-1-one, scytalone (5), line 173, it does not make sense to place the wedge on the hydrogen at C-4 and not show the absolute configuration of the C-3.
Correct the product error of the last decarboxylation step, Figure 1, line 211.The correct diagram is shown:
J. Nat. Prod. 2016, 79, 3, 523–530.
Some molecules must be corrected in bond angles, for example, example: 26, line 271; pestiocandin 70, line 451; 74, line 499:
Check the structure of pestalothioprolides E and F (80, 81), it is not “=R”, change by “-R”, line 526; correct structure 196, the carbon must bond to the oxygen atom, line 1185.
Typos, where the names are superimposed on the chemical structure, example: 110, line 680, 127, line 761.
Although some biological activities are mentioned, there is no explanation of the structure with the reported activity, if possible, it would be convenient to mention something regarding the relationship structure activity.

Author Response
To Editors and Reviewers
I am so sorry for delaying the revised submission, because our city was quarantined for about 10 days, I could not go to my office and deal with the manuscript. Before the quarantine, we had finished most revision, only structures of two molecules left. This morning I went to the campus, and asked my student to carry out my computer and brought it back to my house, because I was still not allowed to come into the campus.
According to opinions from reviewers, we revised the manuscript. In addition, we found some other errors and corrected them.
Below is the responses to Reviewer 2
The authors describe the origin of the compounds isolated from the Pestalotiopsis genus, their biological importance, chemical structures and even some metabolite synthesis. It is an exhaustive and interesting revision, however there are some errors that must be taken care of, and they are mentioned below:
Stereochemical descriptors should be used appropriately (R,S, cis, trans, Z, E), Table 1, and the full manuscript.
Response:
We checked the names and structures of the metabolites and revised all the stereochemical descriptors all around the manuscript.
Check carefully all the structures and names of the molecules, for example naphthalen-1-one, scytalone (5), line 173, it does not make sense to place the wedge on the hydrogen at C-4 and not show the absolute configuration of the C-3.
Responses
- All the names and structures of the metabolites were copied and pasted from the cited articles. In addition, all the structures were strictly re-illustrated in order to make these structures correct as the original structures presented in the original articles.
About the structure of scytalone (5), we checked and revised it according the article (Bunyapaiboonsri et al., 2021)
Bunyapaiboonsri T, Yoiprommarat S, Nithithanasilp S, Choowong W, Preedanon S, Suetrong S. 2021. Two new farnesyl hydroquinones from Pestalotiopsis diploclisia (BCC 35283), the fungus associated with algae. Nat Prod Res, 30:1-7. https://doi.org/10.1080/14786419.2021.1946536
Correct the product error of the last decarboxylation step, Figure 1, line 182.The correct diagram is shown:
J. Nat. Prod. 2016, 79, 3, 523–530.
Responses:
We checked the figure against the preliminary article (J. Nat. Prod. 2016, 79, 3, 523–530), and revised the errors in the manuscript, as shown in the text.
Some molecules must be corrected in bond angles, for example, example: 26, line 255; pestiocandin 70, line 430; 74, line 481:
Responses:
All the molecules were re-illustrated according to the cited articles, and we did not change the bond angles.
Check the structure of pestalothioprolides E and F (80, 81), it is not “=R”, change by “-R”, line 497; correct structure 196, the carbon must bond to the oxygen atom, line 1160.
Responses:
- We carefully checked the structures of pestalothioprolides E and F (80, 81) against the preliminary article (J Nat Prod, 79(9):2332-2340), their structures were revised
- We changed the structure of 196.
Typos, where the names are superimposed on the chemical structure, example: 110 line 640,127, line 734.
Responses:
We regulated the positions of their names.
Although some biological activities are mentioned, there is no explanation of the structure with the reported activity, if possible, it would be convenient to mention something regarding the relationship structure activity.
Responses:
The manuscript is involved in many metabolites, if we do according to the opinion of the reviewer, the manuscript will be too long. In addition, in the cited articles, authors explained the relationships between structures and activity.
Round 2
Reviewer 2 Report
Please, is necessary to correct the structures bond angles, for example: 26, 70 and 74 compounds.
Author Response
I cannot understand what is the SI, and I cannot do anything about it. More important, according to the opinions from the editior, I checked all the manuscript and did my best to improve it. I found some spelling and gramma rerrors and questions about cited references.